# Diallel Analysis: Choosing Parents to Introduce New Variability in a Recurrent Selection Population

**Paulo Henrique Ramos Guimarães [1],\*** , **Adriano Pereira de Castro [2]**, **José Manoel Colombari Filho [2]**, **Paula Pereira Torga [2]**, **Paulo Hideo Nakano Rangel [2]** and **Patrícia Guimarães Santos Melo [1]**

1. Escola de Agronomia, Universidade Federal de Goiás, Rodovia GO-462, km 0, Campus Samambaia, Goiânia 74001-970, GO, Brazil; pgsantos@gmail.com
2. Embrapa Arroz e Feijão, Rodovia GO-462, km 12, Santo Antônio de Goiás 75375-000, GO, Brazil; adriano.castro@embrapa.br (A.P.d.C.); jose.colombari@embrapa.br (J.M.C.F.); paula.torga@embrapa.br (P.P.T.); paulo.hideo@embrapa.br (P.H.N.R.)
\* Correspondence: paulohenriquerg@hotmail.com

**Abstract:** Selecting appropriate donors and acquiring information about the genetic basis of inheritance is essential for breeding programs. In this study, a diallel cross was produced by crossing 15 progenies with five commercial lines of wide diversity for different rice traits (grain yield, plant height, days to flowering, panicle blast, brown spots, leaf scald, and grain discoloration) in an incomplete crossing design. The 20 parents and the 25 $F_2$ crosses constituting the diallel cross were evaluated in a triple lattice design for different traits in a field test. The analysis of variance revealed significant differences between parents and their crosses for all traits, showing high variability. The general combining ability (*GCA*) and the specific combining ability (*SCA*) were significant, with a greater contribution of the *SCA* compared to *GCA* for the variation among crosses, indicating that non-additive effects were more prevalent for the traits evaluated. The results suggested that commercial lines such as IRGA 424 and BRS Catiana can be used to improve CNA 12T population.

**Keywords:** rice; Griffing's method; additive and non-additive gene effects; genetic parameters

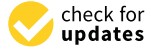



## 1. Background

As one of the most important food crops, rice is cultivated worldwide and is the primary food source for more than half of the global population [1,2]. In this scenario, maintaining genetic variability is essential for developing superior cultivars in order to supply the ever-growing rice demand. As a result, crop breeding includes creating genetic variability and using it to generate improved cultivars [3]. However, its success depends on the amount of genetic variability available for exploitation and the extent to which desirable traits are inherited [4,5]. Therefore, recurrent selection is a breeding method that maintains genetic variability and increases the likelihood of choosing superior lines with continuous selection gains. This cyclical method concentrates desirable alleles by selecting the best progenies in a population [6–8].

The first step in establishing a recurrent selection population is developing the base population, which must have high genetic variability. Therefore, breeders need to choose parent groups that are adapted and genetically complementary [9]. In order to attain this purpose and select the best parents, breeders can use the diallel analysis method. This method indicates the best parents for hybridization and provides the best genetic complementation for target traits [8,10]. Moreover, this method allows the study of inheritance and genetic control related to quantitative traits [11,12].

Furthermore, recurrent selection enables the addition of new genotypes to increase genetic variability in the population during the recombination phase. This procedure can be necessary when the genetic variability is exhausted due to the repeated application of this scheme in small populations, reducing the gains obtained by selection [13]. Therefore,

efficiently identifying parents for crossing allows for selecting the best hybrid combinations, increasing the likelihood of obtaining a promising segregating population. In this scenario, information about the general (*GCA*) and specific combining abilities (*SCA*) coupled with genetic parameters are helpful tools for choosing the better selection strategy and selecting appropriate parents for crosses [14–16].

There are several methodologies to perform diallel analysis [17–21]. In rice, several studies [22–26] have used this technique to estimate the *GCA* and *SCA* for various agronomic traits. However, although plant breeders frequently use combination studies to analyze new donors and their usefulness in crosses [27], studies on the application of diallel analysis in the reintroduction of genetic variability in recurrent selection populations are scarce or nonexistent. Nevertheless, according to [24], there are important questions and some restrictions about using diallel analysis as a source of information that can contribute to breeding programs. From this perspective, the main objectives of this study were: (*i*) evaluating the genetic potential of five rice cultivars as new allele donors for the CNA 12T population; (*ii*) determining the combining ability of fifteen progenies of CNA 12T; and (*iii*) determining the nature of gene action and the magnitude of associated genetic components for different traits in rice.

## 2. Material and Methods

### 2.1. Genetic Material

The genetic material consisted of 25 $F_2$ progenies derived from crosses between two parental groups (Table 1). The first group (grain type "long and slender"), representing male parents, consisted of five commercial cultivars with the ideotype sought by Brazilian rice breeders. These commercial cultivars were chosen based on their varied morphological features, especially high grain yield, earliness, and rice blast resistance. The second group, representing the female parents, consisted of 15 $S_{0:3}$ progenies of the second cycle of the CNA 12T population of lowland rice. This population was synthesized in 2002 by the rice breeding program developed at the Brazilian Agricultural Research Corporation (Embrapa Rice and Beans) as a genetically broad-based population with stable genetic resistance to *Magnaporthe oryzae* (anamorph *Pyricularia grisea* Sacc.) [28,29]. Multi-location trials were used to select 15 $S_{0:3}$ progenies of the CNA 12T population. This selection was based on high heritability traits such as plant height (PH), days to flowering (DTF), disease resistance, grain shape, and lodging resistance. Crosses between the two groups were performed by hand pollination in a greenhouse at Embrapa Rice and Beans, located in Santo Antônio de Goiás, Brazil (16°27′28″ S, 49°19′52″ W, at an elevation of 823 m above sea level). Since reciprocal crosses were not performed, 15 $S_{0:3}$ progenies were established as female parents and five commercial lines were the male donors. Each progeny was sown in excess to ensure the ultimate presence of one plant per pot. The plants were thinned to one per plot twelve days after sowing. Before anthesis, the female parents (15 $S_{0:3}$ progenies) had their anthers removed using a small vacuum pump to extract the unripe anthers from the spikelets before being pollinated with a mixture of pollen from several plants of the same commercial line. After pollination, each panicle was protected with a paper bag to avoid contamination.

### 2.2. Experimental Conditions

The $F_1$ crosses were self-pollinated to obtain the 25 $F_2$ crosses. Along with the 20 parents and 4 checks (BRS Tropical, BRS Jaçanã, Epagri 109, and elite line BRA051108), these crosses were evaluated at the experimental station of Embrapa Rice and Beans, located in Goianira, GO, Brazil (16°26′46″ S, 49°24′22″ W, 732 m). The trial was conducted in a lowland area with continuous flooding until grain maturity and following a triple lattice design. Each plot consisted of five rows of five meters, a row space of 0.17 m, and 60 manually sown seeds per meter. Fertilization consisted of 500 kg ha$^{-1}$ of 04-28-20 (NPK) + Zn. Topdressing fertilization was performed using 150 kg ha$^{-1}$ of urea two times: 15 days after sowing (DAS) and 45 DAS (75 kg ha$^{-1}$ N in each). The technical itinerary for crop management was similar

to commercial production, except for fungicide application. In our case, fungal diseases were not chemically controlled since disease tolerance was part of the experimental assessment. On the other hand, weeds and insect pests were controlled by mechanized spraying when needed.

Seven traits were measured: grain yield (GY, kg ha$^{-1}$), measured in the useful area, the two central rows (1.36 m$^2$), by harvesting all the grains of each plot, which were then dried to 13% moisture; plant height (PH, cm), measured from the ground to the tip of the main tiller; days to flowering (DTF, days), determined as the number of days from sowing to 50% of plants at anthesis; panicle blast (Pb, scale); brown spots (BP, scale); leaf scald (Ls, scale); and grain discoloration (Gd, scale). A visual evaluation considered the percentage of panicles, leaf area, and grains affected by blast, scald, and spots. Grades were attributed according to a visual diagrammatic scale ranging from 0 to 9 (0: no incidence; 1: 1 to 5%; 3: 6 to 12%; 5: 13 to 25%; 7: 26 to 50%; and 9: >50% infection) as proposed by [30].

**Table 1.** Schematic design used in the experiment and crosses between the parents of groups 1 (male parents—commercial rice lines) and 2 (female parents—CNA 12T).

| CNA 12T Progenies | Commercial Rice Lines | | | | |
|---|---|---|---|---|---|
| | **1': IRGA 424** | **2': BRS Biguá** | **3': BRS Catiana** | **4': Federarroz 50** | **5': Epagri 106** |
| 1: CNAx16209-10-B-B-B | - | X | - | - | - |
| 2: CNAx16210-19-B-B-B | X | - | - | - | X |
| 3: CNAx16211-14-B-B-B | - | - | - | X | - |
| 4: CNAx16217-1-B-B-B | X | X | - | - | - |
| 5: CNAx16219-2-B-B-B | - | X | - | - | - |
| 6: CNAx16219-22-B-B-B | X | - | - | - | X |
| 7: CNAx16220-9-B-B-B | X | - | - | - | - |
| 8: CNAx16221-6-B-B-B | X | X | - | - | X |
| 9: CNAx16222-11-B-B-B | X | X | - | - | - |
| 10: CNAx16222-20-B-B-B | - | - | X | X | - |
| 11: CNAx16223-5-B-B-B | - | - | - | X | X |
| 12: CNAx16224-20-B-B-B | - | - | X | X | - |
| 13: CNAx16224-5-B-B-B | - | X | - | - | - |
| 14: CNAx16225-1-B-B-B | - | - | - | X | - |
| 15: CNAx16225-17-B-B | - | - | - | X | X |

### 2.3. Statistical Data Analysis

The experimental design was a 7 × 7 triple lattice design with the genotypes (crosses, parents, and checks) serving as factors. First, the analysis of variance (ANOVA) was performed to determine whether or not the ANOVA assumptions were met. After these preliminary evaluations, the analysis of variance was performed for all the traits according to the fixed linear model, as given below:

$$Y_{ijkm} = \mu + r_j + b_{k/j} + t_m + g_{i/m} + e_{ijkm}$$

where:

$Y_{ijkm}$: is the observed value of the *i*th genotype, in the *k*th block, in the *j*th replicate, belonging to type *m*.

$\mu$: is the constant inherent to all observations.

$r_j$: is the effect of the *j*th replicate, $j = 1, 2, \ldots, J$.

$b_{k/j}$: is the effect of the *k*th block, *k* ($k = 1, 2, \ldots, K$), within the *j*th replicate, $k = 1, 2, \ldots, K$.

$t_m$: is the effect of group *m* in eight groups: $GRP_1$—parental group of commercial rice cultivars; $GRP_2$—parental group of 15 progenies of the CNA 12T population; $GRP_3$—check cultivar; $GRP_4$—crosses with IRGA 424; $GRP_5$—crosses with BRS Biguá; $GRP_6$—crosses with BRS Catiana; $GRP_7$—crosses with Federarroz 50; and $GRP_8$—crosses with Epagri 106;

$g_{i/m}$: is the effect of the *i*th genotype within group *m*;

$e_{ijkm}$: is the average experimental error associated with the *ijkm*th plot, assuming $I.I.D \cap (0, \sigma^2)$, where $I.I.D$ stands for independent and identically distributed.

The genotypes were treated as of fixed effect since the groups included lines or cultivars selected for desirable traits in the breeding program, i.e., they were not randomly selected. Then, the following genetic parameters were estimated for each genotype group based on the mathematical expectation of the mean squares: ($GRP_1$ to $GRP_8$): selective accuracy ($\hat{r}_{\hat{g}g}$), using $\hat{r}_{\hat{g}g} = (1 - 1/F)^{1/2}$, where *F*: is the value of the *F* test (from Snedecor) for each source of variation [31]; quadratic component of the genotype group ($\phi_{grp1}$, $\phi_{grp2}$, $\phi_{grp3}$, ... , $\phi_{grpn}$), estimated from the expressions of the expected mean squares ($\phi_{grp1} = \sigma^2 + k_1 \phi_{grp1}$, $\phi_{grp2} = \sigma^2 + k_2 \phi_{grp2}$, ... , $\phi_{grpn} = \sigma^2 + k_n \phi_{grpn}$); coefficient of genetic determination ($H^2$), equivalent to broad-sense heritability when the genotypes correspond to random effects using the equation $H^2 = \phi_{grpn}/(MS_{en}/k + \phi_{grpn})$, where $\phi_{grpn}$: is the quadratic component of the respective genotype group ($\phi_{grp1}$, $\phi_{grp2}$, ... , $\phi_{grpn}$), $MS_{en}$ is the residual mean square of the ANOVA and *k* is the coefficient associated with ($\phi_{grpn}$) in the expected mean square values of the respective genotype group; and the relative coefficient of variation ($CV_r$), via $CV_r = CV_g/CV$, where $CV_g$ is the genetic coefficient of variation and *CV* is the experimental coefficient of variation [32].

In the presence of balanced data, the *k* coefficient is a direct function of the number of replicates. However, as the level of unbalanced data increases, the *k* value departs from this relationship. Because of the unbalanced experimental data, we decided to estimate the *k* coefficient for the eight genotype groups using the general equation of [33]: $k = \frac{1}{n-1}tr(W^{-1}) + \frac{1}{n}\sum(W^{-1})$; where *n* is the number of genotypes in the respective group and *tr* is the trace of $W^{-1}$, corresponding to the core matrix from the expression of the sum of squares (*SS*), where $SS = \hat{\beta}'W^{-1}\hat{\beta}$ and $\hat{\beta}$ is the solution vector for the genotype group. In this case, *SS* can be estimated by the general equation of the *SS* hypothesis using linear models [34]. Thus, $SS = (C\hat{\beta})'(CQC')^{-1}C\hat{\beta}$, which, by deduction, compares $\beta'W^{-1}\hat{\beta}$ to $\hat{\beta}'C'(CQC')^{-1}C\hat{\beta}$, then $W^{-1} = C'(CQC')^{-1}C$. In these expressions, *C* is the matrix of contrasts between the estimates of the effect vector $\hat{\beta}$ and *Q* can be obtained from the covariance matrix of vector $\hat{\beta}$ divided by the mean square error [35].

The *ggplot2* package was used to visualize the variability between the different genotype groups [36]. The different group means and the genotypes within groups were tested by the Tukey test ($p \leq 0.05$). The analysis was carried out with the R software [37].

## 2.4. Diallel Analysis

Diallel analysis for the general combining ability (*GCA*) and specific combining ability (*SCA*) followed Griffing's method 2 [18]. The general linear model was adapted for the data available for each trait according to procedures given by [35,38], which required diagonalizing the inverse error covariance matrix ($V^{-1}$) by Cholesky's factorization [39]. This procedure provided the congruent matrix U, and $U'U = V^{-1}$ allowed for estimating all parameters by simplifying the ordinary Gauss–Markov structure [38]. The design matrix used the following restrictions: $\sum_{i}^{p}\hat{g}_i = 0$; $\sum_{j}^{p}\hat{g}_j = 0$; $\sum_{i}^{p}\hat{s}_{ij} = 0$ for each *j*, and $\sum_{j}^{p}\hat{s}_{ij} = 0$ for each *i* [21]. The significance of the estimates of the $GCA_1$, $GCA_2$, and *SCA* effects was evaluated by the *t*-test ($p \leq 0.05$). The 95% confidence interval and the standard deviation of each estimate were obtained from the square root of their corresponding variance in the diagonal of the covariance matrix $\hat{V}(\hat{\beta})$.

The quadratic components associated with the *GCA* of the two groups and the *SCA* effects were estimated by the method of moments based on the mean square expectation, as follows: $\phi_{GCA_1} = \frac{MS_{GCA_1} - MS_e}{JKL}$; $\phi_{GCA_2} = \frac{MS_{GCA_2} - MS_e}{IKL}$; $\phi_{SCA} = \frac{MS_{SCA} - MS_e}{K}$; where *I* is the number of parents of group 1, *J* is the number of parents of group 2, *K* is the number of replicates, $MS_{GCA_1}$ and $MS_{GCA_2}$ are the mean squares of the *GCA* effects of groups 1 and 2, respectively, $MS_{SCA}$: is the mean square of the *SCA* effect, and $MS_e$: is the mean square error. The relative importance of the *GCA* for both groups and the *SCA*

in determining the genotypic performance of crosses was assessed by Baker's ratio [40]:
$\dfrac{(2\phi_{GCA_1} + 2\phi_{GCA_2})}{(2\phi_{GCA_1} + 2\phi_{GCA_2} + \phi_{SCA})}$; where $\phi_{GCA_1}$, $\phi_{GCA_2}$, and $\phi_{SCA}$ are quadratic components associated with the $GCA_1$, $GCA_2$ and $SCA$ effects. All necessary matrix operations to implement the general linear model and thus obtain the parameter estimates and their associated errors were performed using the R platform [37].

## 3. Results

### 3.1. Phenotyping Screening and Genetic and Variation Parameters

A summary of the results obtained for the overall shoot phenotyping and ANOVA is presented in Figure 1. As expected, grain yield (GY), plant height (PH), days to flowering (DTF), panicle blast (Pb), brown spots (Bs), leaf scald (Ls), and grain discoloration (Gd) showed high variability, with CVs ranging from 1.91% (PH) to 20.03% (Gd) (Figure 1A–G). The magnitude of the mean squares of the genotypes (crosses and parents) indicated significant differences ($p \leq 0.01$) between genotypes for all traits, suggesting the presence of genetic variability. For some traits, e.g., DTF and Ls, the mean squares owing to cross effects ($GRP_4$ to $GRP_8$) were highly significant. Except for GY and Bs, the two parent groups were statistically different between and within them. In general, $GRP_2$ was more productive, taller, and more sensitive to diseases (Pb, Bs, and Ls) (Figure 1A,B,D–F) than $GRP_1$, but statistically similar to the first group, highlighting the potential of the CNA 12T population to generate superior inbreds.

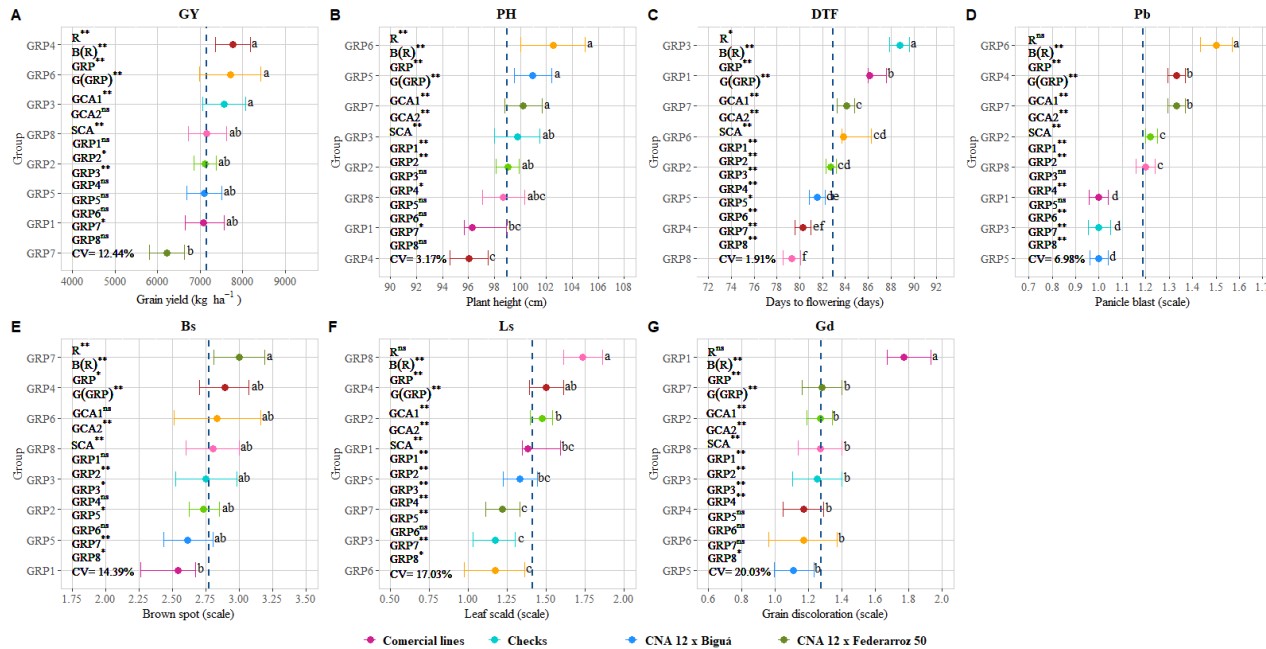

**Figure 1.** Variation in the means and 95% confidence intervals for different rice traits. (**A**) grain yield (GY), (**B**) plant height (PH), (**C**) days to flowering (DTF), (**D**) panicle blast (Pb),(**E**) brown spots (BP), (**F**) leaf scald (Ls) and (**G**) grain discoloration (Gd). [ns] Non-significant; * and ** significant by the *t*-test at the 0.05 and 0.01 probability levels, respectively; R: replicates; B(R): block within replicates; GRP: group; $GCA_1$: general combining ability of the parental group of commercial rice cultivars; $GCA_2$: general combining ability of the parental group of progenies of the CNA 12T population; SCA: specific combining ability; $GRP_1$: parental group of commercial rice cultivars; $GRP_2$: parental group of progenies of the CNA 12T population; $GRP_3$: checks; $GRP_4$: group of crosses with IRGA 424; $GRP_5$: group of crosses with BRS Biguá; $GRP_6$: group of crosses with BRS Catiana; $GRP_7$: group of crosses with Federarroz 50; $GRP_8$: group of crosses with Epagri 106; CV: coefficient of variation. Means followed by the same letter do not differ statistically according to the Tukey test (*p*-value 0.05). Dashed lines represent the mean (progenies, checks, parents, and crosses) of each measured trait.

In the case of hybrids, we found that the crosses with IRGA 424 ($GRP_4$) were more productive, earlier, and with lower plant height than the parental groups ($GRP_1$ and $GRP_2$). The crosses with Epagri 106 ($GRP_8$) showed the same results (Figure 1A–C). However, although the hybrids derived from the crosses with BRS Biguá ($GRP_5$) had lower GY than $GRP_4$ and $GRP_8$, the progenies of $GRP_5$ did not differ from the $GRP_4$ hybrids (Figure 1A) and had low disease incidence, especially Pb (Figure 1D) and Gd (Figure 1G). Furthermore, the progenies derived from this cross did not differ statistically from the parental group and checks for GY (Figure 1A). These results showed that the genotypes evaluated had significant diversity, and this genetic variability can be exploited through selection by rice breeding programs.

The summary of the genetic parameters (selective accuracy ($\hat{r}_{\hat{g}g}$), coefficient of genetic determination ($H^2$) and coefficient of relative variation ($CV_r$)) is given in Table 2. Except for Pb, Bs, and Gd in $GRP_3$, $GRP_6$, and $GRP_5$, respectively, the genetic parameters were not estimated since the quadratic component was negative. The selective accuracy ($\hat{r}_{\hat{g}g}$) varied greatly between evaluated traits, ranging from 23.5% (Bs) to 99.8% (Pb). All traits had at least one group with low to moderate $\hat{r}_{\hat{g}g}$ estimates [31]. Despite these results, traits such as DTF, Pb, Ls, and Gd had high experimental precision ($\hat{r}_{\hat{g}g} > 70\%$), indicating that the estimated values are valid compared to real values, providing more statistical reliability in the estimates of target traits [31,41].

**Table 2.** Estimates of genetic variance parameters for different traits in eight genotype groups derived from the crosses between the CNA 12T population and commercial rice lines.

| Groups | $\hat{r}_{\hat{g}g}$ | $H^2$ | $CV_r$ | $\hat{r}_{\hat{g}g}$ | $H^2$ | $CV_r$ |
|---|---|---|---|---|---|---|
| | | GY | | | PH | |
| $GRP_1$ | 0.296 | 0.087 | 0.140 | 0.927 | 0.859 | 1.112 |
| $GRP_2$ | 0.713 | 0.508 | 0.454 | 0.906 | 0.821 | 0.003 |
| $GRP_3$ | 0.893 | 0.798 | 0.833 | 0.397 | 0.157 | 0.001 |
| $GRP_4$ | 0.466 | 0.217 | 0.223 | 0.766 | 0.586 | 0.002 |
| $GRP_5$ | 0.268 | 0.072 | 0.118 | 0.508 | 0.258 | 0.001 |
| $GRP_6$ | 0.369 | 0.136 | 0.164 | 0.844 | 0.713 | 0.002 |
| $GRP_7$ | 0.769 | 0.591 | 0.509 | 0.766 | 0.587 | 0.002 |
| $GRP_8$ | 0.637 | 0.405 | 0.360 | 0.454 | 0.206 | 0.001 |
| | | DTF | | | Pb | |
| $GRP_1$ | 0.976 | 0.953 | 2.121 | 0.916 | 0.840 | 1.032 |
| $GRP_2$ | 0.955 | 0.912 | 1.442 | 0.992 | 0.984 | 3.524 |
| $GRP_3$ | 0.956 | 0.914 | 1.363 | † | † | † |
| $GRP_4$ | 0.949 | 0.901 | 1.281 | 0.995 | 0.990 | 4.226 |
| $GRP_5$ | 0.833 | 0.695 | 0.640 | 0.368 | 0.136 | 0.168 |
| $GRP_6$ | 0.991 | 0.982 | 3.313 | 0.998 | 0.996 | 6.675 |
| $GRP_7$ | 0.932 | 0.868 | 1.088 | 0.994 | 0.988 | 3.807 |
| $GRP_8$ | 0.565 | 0.319 | 0.288 | 0.992 | 0.984 | 3.331 |
| | | Bs | | | Ls | |
| $GRP_1$ | 0.608 | 0.370 | 0.345 | 0.939 | 0.882 | 1.232 |
| $GRP_2$ | 0.874 | 0.764 | 0.805 | 0.960 | 0.921 | 1.525 |
| $GRP_3$ | 0.851 | 0.725 | 0.681 | 0.882 | 0.779 | 0.786 |
| $GRP_4$ | 0.235 | 0.055 | 0.102 | 0.961 | 0.923 | 1.472 |
| $GRP_5$ | 0.798 | 0.637 | 0.561 | 0.961 | 0.924 | 1.476 |
| $GRP_6$ | † | † | † | 0.451 | 0.204 | 0.210 |
| $GRP_7$ | 0.899 | 0.809 | 0.871 | 0.929 | 0.862 | 1.062 |
| $GRP_8$ | 0.788 | 0.621 | 0.539 | 0.815 | 0.663 | 0.592 |

**Table 2.** *Cont.*

| Groups | $\hat{r}_{\hat{g}g}$ | $H^2$ | $CV_r$ | $r_{\hat{g}g}$ | $H^2$ | $CV_r$ |
|---|---|---|---|---|---|---|
| | GY | | | PH | | |
| | Gd | | | | | |
| $GRP_1$ | 0.933 | 0.870 | 1.168 | | | |
| $GRP_2$ | 0.935 | 0.874 | 1.176 | | | |
| $GRP_3$ | 0.955 | 0.913 | 1.355 | | | |
| $GRP_4$ | 0.926 | 0.857 | 1.038 | | | |
| $GRP_5$ | † | † | † | | | |
| $GRP_6$ | 0.836 | 0.699 | 0.632 | | | |
| $GRP_7$ | 0.394 | 0.155 | 0.182 | | | |
| $GRP_8$ | 0.811 | 0.658 | 0.584 | | | |

$\hat{r}_{\hat{g}g}$: selective accuracy; $H^2$: coefficient of genetic determination; $CV_r$: coefficient of relative variation from ANOVA; GY: grain yield; PH: plant height; DTF: days to flowering; Pb: panicle blast; Bs: brown spot; Ls: leaf scald; Gd: grain discoloration; $GRP_1$: parental group of commercial rice cultivars; $GRP_2$: parental group of progenies of the CNA 12T population; $GRP_3$: checks; $GRP_4$: group of crosses with IRGA 424; $GRP_5$: group of crosses with BRS Biguá; $GRP_6$: group of crosses with BRS Catiana; $GRP_7$: group of crosses with Federarroz 50; $GRP_8$: group of crosses with Epagri 106. †: φ < 0: negative estimate for the quadratic component of the group.

The square root of the estimates of the coefficient of genetic determination ($H^2$) reflects $\hat{r}_{\hat{g}g}$, which indicates the precision in predicting genetic values, referring to the correlation between predicted genetic values and the actual genetic values of the genotypes [31,42]. The estimates found for $H^2$ in some groups showed low values ($H^2 < 30\%$), as found for GY ($GRP_1$, $GRP_4$, and $GRP_6$), PH ($GRP_3$, $GRP_5$, and $GRP_8$), Bs ($GRP_1$ and $GRP_4$), and Gd ($GRP_1$) (Table 2). However, for some groups of these traits and for DTF, Pb, Ls, and Gd, the $H^2$ estimates can be considered high ($H^2 > 70\%$) [43]. Therefore, the genetic values predicted for the different groups are reliable due to the high $\hat{r}_{\hat{g}g}$ estimates.

The relative variation coefficient ($CV_r$) varied greatly according to the trait and phenotypic group, ranging from 0.001 (PH) to 6.675 (Pb) (Table 2). Unsurprisingly, the values provided by $CV_r$ were higher than 1, especially for DTF, Pb, Ls, and Gd. The higher the $CV_r$ value, the greater the genetic control of the trait and the lower the influence of environmental factors on phenotypic performance [44]. Moreover, $CV_r > 1.0$ indicates a favorable situation for selecting superior genotypes, especially among the progenies derived from the crosses performed [29,42].

### 3.2. Diallel Analysis and Genetic Components of Variance

Figure 1 shows the partitioning of the overall sum of squares (*GCA*) and the specific (*SCA*) combining ability. Except for GY ($GCA_2$) and Bs ($GCA_1$), the *GCA* effect was highly significant ($p \leq 0.01$) for both parental groups ($GRP_1$—commercial rice cultivars and $GRP_2$—progenies of the CNA 12T population). These findings indicate that at least one parent was superior to the others with regard to the mean performance of its hybrid combinations. This result can be due to differences between the parents in their ability to transmit additive effect alleles, thus influencing the behavior of the hybrids for the evaluated traits [14,45]. Furthermore, the *SCA* effect was also highly significant ($p \leq 0.01$), showing that not only additive effects but also non-additive gene action was involved in the genetic control of the evaluated traits, thus indicating differences between parent groups and their crosses regarding *GCA* and *SCA* effects in the $F_2$ generation.

The comparative estimates of quadratic components due to *GCA* and *SCA* effects revealed the importance of *SCA* in controlling the evaluated traits since the quadratic component of *SCA* ($\phi_{SCA}$) was higher than *GCA* ($\phi_{GCA_1}$ and $\phi_{GCA_2}$) for most traits (Table 3). The higher *SCA* magnitude suggests a significant role of non-additive gene action such as dominance and epistatic effects in controlling the evaluated traits. These results can be supported by Baker's ratio, which ranged from 0.193 (Pb) to 0.720 (PH) (Table 3). If the values are lower than one, they suggest that *SCA* can predict the performance of crosses.

**Table 3.** Quadratic component estimates for the general ($\phi_{GCA_1}$ and $\phi_{GCA_2}$) and specific ($\phi_{SCA}$) combining abilities of different traits in two groups of rice.

| Quadratic Components | | GY | PH | DTF | Pb | Bs | Ls | Gd |
|---|---|---|---|---|---|---|---|---|
| $\phi_{GCA_1}$ | | 135,841.17 | 6.654 | 5.296 | 0.011 | 0.0131 | 0.039 | 0.009 |
| $\phi_{GCA_2}$ | | 12,542.51 | 0.794 | 0.613 | 0.008 | 0.0134 | 0.015 | 0.011 |
| $\phi_{SCA}$ | | 293,113.85 | 5.781 | 7.157 | 0.159 | 0.077 | 0.128 | 0.069 |
| | Baker's ratio | 0.503 | 0.720 | 0.623 | 0.193 | 0.408 | 0.458 | 0.362 |

GY: grain yield; PH: plant height; DTF: days to flowering; Pb: panicle blast; Bs: brown spot; Ls: leaf scald; Gd: grain discoloration.

### 3.3. Breeding Potential of the Parents Estimated by GCA

The magnitude and direction of *GCA* effects, as shown in Figure 2, can provide guidelines for selecting parents and allowing their use by breeders [46,47]. The *GCA* effects revealed that none of the parents were good combiners for all traits measured simultaneously. The *GCA* estimate showed that the best combining parent to improve the GY was CNAx16225-17-B-B-B (Figure 2A). Besides, we found a low, negative, and highly significant *GCA* estimate ($p \leq 0.01$) for Pb (Figure 2D), indicating that this parent could increase the resistance to panicle blast. On the other hand, using CNAx16225-17-B-B-B in hybridization can increase rice susceptibility to Bs, Ls, and Gd (Figure 2E–G).

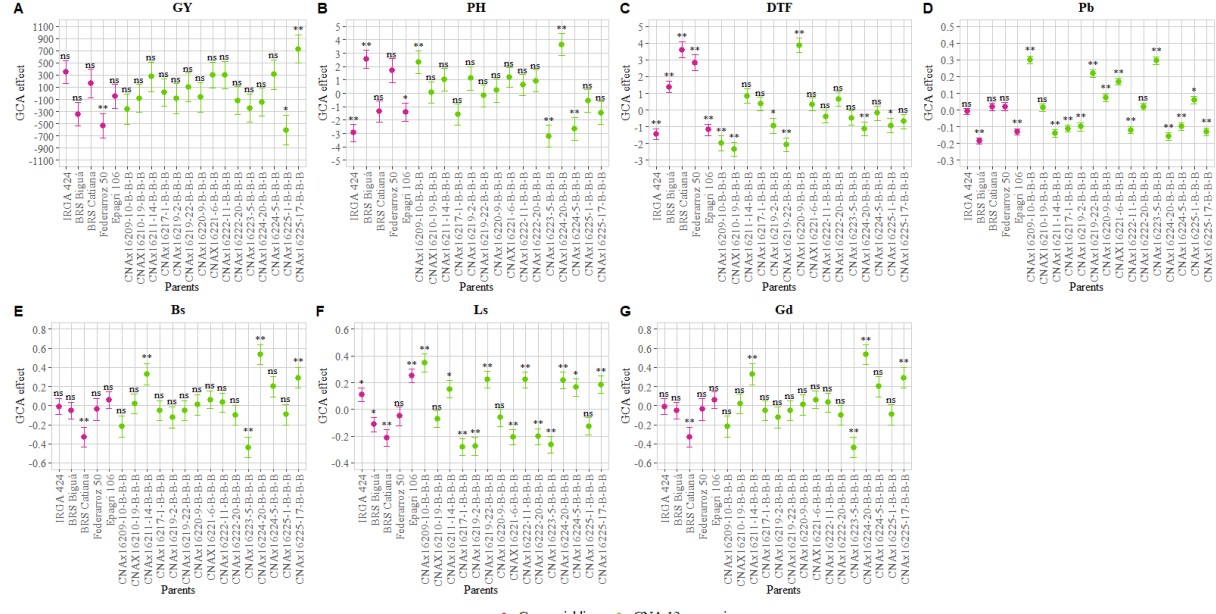

**Figure 2.** Estimates of the general combining ability effects (*GCA*) for two groups (five parents of group 1 and 15 parents of group 2) of rice parents. (**A**) grain yield (GY), (**B**) plant height (PH), (**C**) days to flowering (DTF), (**D**) panicle blast (Pb), (**E**) brown spots (BP), (**F**) leaf scald (Ls) and (**G**) grain discoloration (Gd). $^{ns}$ Non-significant; * and ** indicate significance at the 0.05 and 0.01 probability levels, respectively, by the *t*-test. Values are *GCA* estimates $\pm$ SD.

The superior general combiners for PH, IRGA 424, and Epagri 106 (Figure 2B) were found to be good general combiners for DTF (Figure 2C), with some advantage for the last genotype, which was a good combiner for Pb (Figure 2D). Low and negative *GCA* values for DTF were found in CNAx16225-1-B-B-B, CNAx16209-10-B-B-B, CNAx16210-19-B-B-B, and CNAx16219-22-B-B-B, indicating a reduction in the crop cycle (Figure 2C). Genotypes CNAx16224-20-B-B-B and CNAx16219-2-B-B-B were identified as good parents for use in hybridization to improve DTF and Pb simultaneously based on the *GCA* effect (Figure 2C,D), with some advantage for the last genotype, which showed good combining ability for Ls (Figure 2F). Besides, CNAx16224-5-B-B-B and CNAx16223-5-B-B-B were

among the best combiners for PH (Figure 2B), providing resistance to Pb (CNAx16224-5-B-B-B and CNAx16223-5-B-B-B), with some advantage for the last genotype, which was a good combiner for Ls and Gd (Figure 2D,F,G).

With regard to the resistance to plant diseases exclusively, the best parents were BRS Biguá (Pb and Ls), BRS Catiana (Bs, Ls, and Gd), CNAx16211-14-B-B-B and CNAx16222-11-B-B-B (Pb), CNAx16217-1-B-B-B (Pb and Ls), and CNAx16221-6-B-B-B and CNAx16222-20-B-B-B (Ls) (Figure 2D–G). Hybridization between these parents should be performed carefully since improving a trait (e.g., Pb) could increase plant susceptibility to Bs and Ls (CNAx16224-20-B-B-B and CNAx16211-14-B-B-B) (Figure 2D–F) and extend the plant cycle (BRS Biguá) (Figure 2C).

In contrast, Federarroz 50 showed a low combining ability with the CNA 12T population based on the negative and highly significant *GCA* estimate ($p \leq 0.01$) for GY (Figure 2A) and the positive and highly significant *GCA* estimate ($p \leq 0.01$) for DTF (Figure 2C). Likewise, CNAx16220-9-B-B-B increased the DTF (Figure 2C) and Pb (Figure 2D), indicating that introducing these parents in the CNA 12T population could reduce its grain yield performance, increase the crop cycle, and favor the incidence of panicle blast.

*3.4. Impact of Crossing CNA 12T and Inbred Lines*

Among the 25 crosses, the 3' × 10 cross (994 Kg ha$^{-1}$) showed positive and highly significant *SCA* effects for GY (Figure 3A). Furthermore, this cross showed negative and significant *SCA* values for Gd (Figure 3G). On the other hand, this hybrid showed positive *SCA* values for PH (Figure 3B), DTF (Figure 3C), Pb (Figure 3D), and Ls (Figure 3F). In contrast, hybrids 1' × 9 and 4' × 10 were among the crosses with the highest negative estimates for GY (Figure 3A), indicating a reduction in grain yield. However, these crosses were among the top hybrids with a negative and highly significant reduction in DTF (Figure 3C), Pb (Figure 3D), and Ls (Figure 3F). Moreover, combinations 2' × 13 and 4' × 12 showed the highest GY reduction (Figure 3A). However, hybrid 2' × 13 showed negative estimates for Ls (Figure 3F) and Gd (Figure 3G). Besides, the 4' × 12 cross showed negative and significant values for PH (Figure 3B) and DTF (Figure 3C).

For DTF, the crosses with negative and significant *SCA* effects were 1' × 8, 3' × 12, 2' × 4, 2' × 9, 5' × 8, and 5' × 15 (Figure 3C), showing that these hybrids were good specific combiners for early rice maturity. Combined with early maturity, these crosses showed a decrease in Pb and Ls (1' × 8 and 3' × 12), Bs and Gd (5' × 15), Bs (2' × 4), Ls (2' × 9), and Pb (5' × 8) (Figure 3D–G). However, hybrid 5'×15 showed a positive and significant value for PH (Figure 3B). Likewise, the 5' × 8 cross increased the Bs, Ls, and Gd (Figure 3E–G). Furthermore, DTF showed positive and significant *SCA* effects for crosses 4' × 14 and 4' × 15, increasing the crop cycle for these hybrids (Figure 3C). However, although increasing the DTF, the hybrid 4' × 15 showed desirable behavior for Ls and Gd (Figure 3F,G).

Resistance to rice diseases such as blast, leaf scald, brown spots, and grain discoloration is desirable in rice breeding [48]. Individually or combined, the desirable crosses with negative and significant *SCA* effects for these traits were 1' × 6, 2' × 1, 5' × 2 and 5' × 6 (Pb), 2' × 8 (Pb and Bs), 4' × 3 (Pb, Ls, and Gd), and 1' × 4 (Pb and Ls) (Figure 3D–G). Despite these results, hybrids 1' × 6, 2' × 1, and 4' × 3 showed positive and significant values for Bs (Figure 3E). Moreover, the 5' × 2 cross had positive and significant *SCA* effects for PH (Figure 3B), showing that this hybrid was not a good specific combiner for plant height reduction. In contrast, crosses 1' × 2 and 1' × 7 had positive and significant values for Pb and Ls (Figure 3D,F). Likewise, hybrids 4' × 11 and 5' × 11 showed an increased incidence of Pb (Figure 3D).

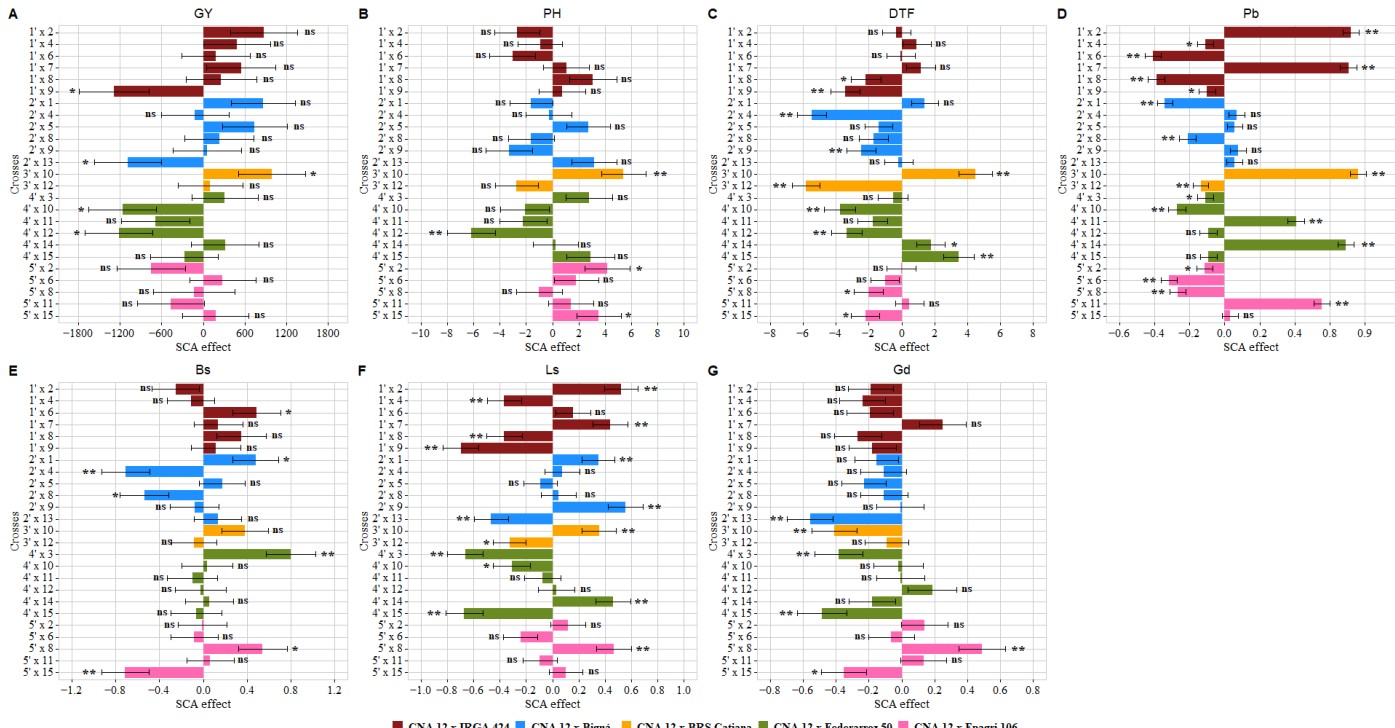

**Figure 3.** Estimates of the specific combining ability effects (*SCA*) for five groups of crosses between two rice parent groups. (**A**) grain yield (GY), (**B**) plant height (PH), (**C**) days to flowering (DTF), (**D**) panicle blast (Pb), (**E**) brown spots (BP), (**F**) leaf scald (Ls) and (**G**) grain discoloration (Gd). [ns] Non-significant; * and ** indicate significance at the 0.05 and 0.01 probability levels, respectively, by the *t*-test. Values are *SCA* estimates ± SD.

## 4. Discussion

The analysis of variance systematically revealed highly significant effects of the genotypes, highlighting differences between crosses, parents, and checks (Figure 1). Moreover, our field phenotyping provided quality data for grain yield (GY), plant height (PH), days to flowering (DTF), panicle blast (Pb), brown spots (Bs), leaf scald (Ls), and grain discoloration (Gd) in lowland rice. Despite high CVs, as found in rice studies by [3,49–51], we found low CVs that confirmed our good performance for rice screening.

Recurrent selection is an open-ended scheme that involves recycling the best genotypes, generation after generation. The cyclic scheme improves the frequency of favorable alleles for different target traits. Moreover, it is possible to obtain improved progenies and add more genetic variability at each selection cycle [6–8]. As a result, some parents may appear more useful than others in rice improvement, depending on the goals of the study. In general, some traits are desirable in rice breeding, e.g., yield potential, grain shape, and grain quality (BRS Catiana, IRGA 424 and Epagri 106) [52,53], tolerance to high nitrogen levels (IRGA 424), plant architecture (BRS Catiana, Epagri 106 and IRGA 424) [52,53], resistance to panicle blast (IRGA 424, BRS Biguá and Fedearroz 50) [54,55], iron toxicity tolerance (IRGA 424 and Epagri 106) [54], and lodging resistance (BRS Catiana) [53].

The CNA 12T population was developed as a source of stable genetic resistance for *Magnaporthe oryzae* (anamorph *Pyricularia grisea* Sacc.) [28,29]. In a previous study performed by [56] with the CNA 12T population, the contribution of environmental effects to phenotypic variance outweighed genetic effects ($\sigma_e^2 > \sigma_g^2$) for panicle blast. This result highlighted an outstanding linear performance of CNA 12T in different environments. Panicle blast is one of the most devastating rice diseases, colonizing leaves, panicles, and other plant parts, resulting in significant losses [57–59]. Panicle blast is a quantitative trait largely influenced by the environment due to the existence of different blast races [60–62]. Therefore, it is essential to identify stable genotypes capable of adapting to a wide range of rice-growing

areas. In this study, by running a diallel analysis, we attempted to analyze the usefulness of a parental rice group (commercial lines) to introduce new genetic variability into the CNA 12T population, especially for Pb, and evaluate the behavior of new populations through these crosses.

It is common to observe predominant additive effects for most traits in early generations of self-pollinated crops [63]. Therefore, we expected to observe polygenic inheritance and predominant additive effects for most traits evaluated. In contrast, we found predominant non-additive effects controlling the expression of individual traits since the $\phi_{SCA}$ effects were higher than the $\phi_{GCA_1}$ and $\phi_{GCA_2}$ effects (Table 3). This finding was confirmed by Baker's ratio, whose values were lower than one. High Baker's ratio values (>0.5), associated with genetic high determination coefficients (Table 2), imply that selection can be effective in early generations [64]. The authors of [27] emphasize that this predominance of non-additive effects offers more opportunities for exploiting hybrid vigor or heterosis in rice improvement. Similar results were found by [64,65] for grain yield and by [14,27] for plant height and days to flowering. Despite the predominance of non-additive effects, as found in the present study, [14] highlighted the presence of considerable additive genetic effects in irrigated rice across Latin America, which can be exploited by breeding programs. Therefore, recurrent selection, such as that used by the rice breeding program at Embrapa, is a valuable method that concentrates positive alleles dispersed among different rice groups and emphasizes the *GCA* effect, in addition to being an efficient method for recombining many genes and increasing the yield potential [66,67].

Originally, the IRGA 424, Epagri 106, and BRS Biguá lines were modern plant types, with low plant height, early plant cycle, and resistance to panicle blast [54,68,69]. The reduction in PH found in the hybrids derived from IRGA 424 and Epagri 106 (Figure 1B) is a consequence of the Green Revolution, which introduced *semidwarf* genes and allowed the selection of genotypes with higher yields and lower plant height [70–72]. Short plants are suitable for high sowing densities, have high tillering capacity, respond to high nitrogen fertilization rates, and lodging resistance, resulting in increased harvest indices [73–75]. Unsurprisingly, the hybrids derived from IRGA 424, Epagri 106, and BRS Biguá showed a net increase in GY (Figure 1A). Moreover, the earliness in maturity found for these progenies (Figure 1C) indicated that they could withstand high temperatures at the end of the growing season, ensuring good grain filling. Coincidentally, IRGA 424, BRS Biguá, and Epagri 106 had high *GCA* effects for PH and DTF, Pb and Ls, and PH, DTF, and Pb, respectively (Figure 2B–D,F). Therefore, these lines contribute favorable alleles for effectively improving the PH, DTF, Pb, and Ls, ultimately improving the grain yield.

According to [76], *GCA* estimates can be used as the breeding value index of a particular genotype. When these values are high, the parental mean predominates over the general mean, indicating the flow of useful genes from parents to offspring at a higher rate, in addition to additive gene action [77]. Therefore, higher *GCA* estimates could indicate higher heritability, reduced environmental effects, and high gene interactions [46,78]. In the present study, where we determined the best combiners among the two groups, we found that none of the parents in either group stood out simultaneously as a good combiner, highlighting the complexity of improving multiple traits and the selection process [79,80]. Herein, negative and significant values were considered for all traits except GY. In this scenario, some parents of the first group ($GRP_1$) showed negative and significant *GCA* values for more than one trait: Bs, Ls, and Gd (BRS Catiana); Pb and Ls (BRS Biguá); PH and DTF (IRGA 424); and PH, DTF, and Pb (Epagri 106). This indicates that this group can be used to improve particular CNA 12T's traits (Figure 2). Similar results were found by [81,82], wherein a single parent was identified as having a favorable *GCA* effect for more than one trait. However, the introduction of these donors to the CNA 12T population must be performed carefully since improving a trait can cause an undesirable increase in other traits (e.g., reduction in Pb (Figure 2B) and increase in PH (Figure 2D), as found for BRS Biguá).

Although there were no significant *GCA* effects found in IRGA 424 and BRS Catiana for GY (Figure 2A), these lines could be directly introduced into the CNA 12T population without harming the productive capacity of the population since they showed positive *GCA* effects for GY. Therefore, these genotypes can result in greater resistance to Pb, Gd, and Bs (IRGA 424 and BRS Catiana) (Figure 2D,E,G), resistance to Ls (BRS Catiana) (Figure 2F), better plant architecture, tolerance to lodging (IRGA 424 and BRS Catiana) (Figure 2B), and earliness (IRGA 424) (Figure 2C). The CNA 12T population, which currently consists of 18 subpopulations [28,29], could be composed of 20 subpopulations, with an additional two coming from the half-sibling families derived from these two new parents.

In contrast, it should be noted that genotype Federarroz 50 is known as an important source of resistance to rice blast [55,83]. However, introducing this parent into the CNA 12T population of the present study reduced its performance in terms of grain yield (Figure 2A) and increased the crop cycle (Figure 2C). In order to overcome this problem, Federarroz 50 could be incorporated into the CNA 12T population via backcrossing using the CNA 12T population as a recurrent parent and selecting resistant progenies in each backcrossing cycle. Through this procedure, the participation of this genotype in the genetic background of the CNA 12T population would be negligible, except for the target alleles of rice blast resistance.

According to [78], the potential of parents to combine well is defined by their ability to transmit and express desirable genes to their progenies. This effect can be noted through desirable *SCA* effects in crosses. Based on this concept, we found two types of gene actions: IRGA 424 × CNAx16222-11-B-B-B and Fedearroz 50 × CNAx16211-14-B-B-B, which had low *GCA* effects for Ls and Gd, respectively (Figure 2F,G). On the other hand, their progenies (1' × 9 and 4' × 3) had high *SCA* effects (Figure 3F,G), which can be explained by over-dominant, non-allelic gene interactions, i.e., of the dominance × dominance type [84]. Another type of gene action could be exemplified for DTF through the crosses between BRS Catiana × CNAx16224-20-B-B-B and Fedearroz 50 × CNAx16224-20-B-B-B. In this case, the desirable additive effects of a good combiner (CNAx16224-20-B-B-B) and the favorable epistatic interactive effects of a poor combiner (BRS Catiana and Fedearroz 50) (Figure 2C) resulted in a high *SCA* effect in the 3' × 12 and 4' × 12 progenies, respectively (Figure 3C). These findings could imply additive × additive gene action [85]. According to [86], this situation is favorable, and these parents can be exploited for rice improvement, resulting in stable transgressive segregants carrying fixable gene effects.

In some cases, e.g., Ls, the 3'×10 hybrid had unfavorable *SCA* effects (Figure 3F). However, the parents used in this combination were good combiners with favorable *GCA* effects (Table 1 and Figure 2F). This result might be linked to the unfavorable allelic combination from the parents, resulting in an undesirable behavior of the target trait. Similar results were found by [11,87], who reported that good general combiners might not produce hybrids with desirable *SCA* values. This finding indicates epistatic gene action, similar to earlier studies, e.g., [88–90], where several traits in rice were found to be controlled by epistatic gene action.

## 5. Conclusions

Parental matching is the key to breeding programs. This study highlighted a wide diversity of different rice traits evaluated in the CNA 12 T population. Our study revealed highly significant differences between parents and their crosses for all traits and the variance was highly significant for most traits studied in the $F_2$ progenies. The results found in this research support [14,66,67], in whose study the recurrent selection method is useful and allows for concentrating positive alleles dispersed among different rice groups. Through this method, general (*GCA*) and specific (*SCA*) combining ability effects can be exploited to improve CNA 12T progenies. However, the relative contribution and changes in *GCA* and *SCA* effects in improving the crosses of commercial lines using the CNA 12T population varied greatly. We also found that the intercrossing of CNA 12T and commercial lines produced hybrids with varying levels of *SCA* effects. Baker's ratio highlighted the greater

importance of non-additive effects governing traits such as grain yield, plant height, days to flowering, panicle blast, brown spots, leaf scald, and grain discoloration. Some donors (e.g., IRGA 424 and BRS Catiana) showed favorable combinations with CNA 12T, highlighting the potential to find desirable transgressions for the genetic improvement of rice. However, it is necessary to evaluate other trials performed in different Brazilian regions in order to compare the measured traits.

**Author Contributions:** A.P.d.C., J.M.C.F., P.H.N.R. and P.P.T. designed the experiment. A.P.d.C., P.P.T. and P.H.R.G. phenotyped the panel. P.H.R.G. performed the statistical analysis. P.H.R.G., A.P.d.C. and P.G.S.M. interpreted the phenotypic results and wrote the paper, which was edited and approved by all co-authors. All authors have read and agreed to the published version of the manuscript.

**Funding:** This research received funding from Embrapa through the rice breeding program for the experiments conducted with lowland rice progenies and from CAPES for the fellowship of the first author.

**Institutional Review Board Statement:** Not applicable.

**Data Availability Statement:** All data generated or analyzed during this study are included in this published article. The phenotypic data are included in Figure 1.

**Acknowledgments:** The authors thank all the Embrapa staff who contributed to the field trial. We also wish to thank the National Council for the Improvement of Higher Education (CAPES) for granting a scholarship to PHRG.

**Conflicts of Interest:** The authors declare that they have no competing interest.

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
