# Peer review of "Diallel Analysis: Choosing Parents to Introduce New Variability in a Recurrent Selection Population"

_agriculture, doi:10.3390/agriculture13071320_

Round 1

Reviewer 1 Report (New Reviewer)

The paper entitled “Diallel analysis: choosing parents to introduce new variability in a recurrent selection population” by Paulo Henrique Ramos Guimarães, Adriano Pereira de Castro, José Manoel Colombari Filho, Paula Pereira Torga, Paulo Hideo Nakano Range and Patrícia Guimarães Santos Melo concerns research related to how new plant varieties are obtained. In this case, the subject of the study is Oryza sativa, commonly referred to as Asian rice. Oryza sativa is a plant species that determines the nutrition of the Earth's population. Achieving high-quality harvests of Oryza sativa is the goal of research by scientists that work on producing new varieties that can have significant advantages for consumers. Therefore, the research presented in this paper is very important. It constitutes another step towards learning about the processes that allow researchers to obtain new varieties of this plant.

The short and compact “Background” is a very good introduction to further parts of the publication.

          In the part of the work entitled “Material and methods”, the authors provided a detailed description of experimental conditions, statistical research methods, as well as diallel analysis. The authors use modern techniques applied in plant breeding processes. The very interesting genetic material that constitutes the basis of the experiment in the presented work draws special attention. Both the materials and methods are described in great detail. Moreover, the methods used in the experiment described in the study do not raise any doubts as to their selection.

A large part of the publication is the "Results" section that contains the detailed results of research. The section describes the data collected that are presented in three figures and supplemented by two tables. The section contains comprehensive and reliable results.

The over two page long “Discussion” is well written and explains the problems that are worth taking into consideration in the work undertaken by authors subject.

The work is well conducted and written and will certainly be interesting for readers, which will translate into its citability. I would not like to go into details about the work because it will repeat the content contained therein.

Again, I would like to underline that the results presented by the researchers in this paper are very important. Finally, I would like also to emphasize that the research concerns future harvests. Improving the quality and size of harvests in the face of climate change and growing population which requires more food are extremely significant.

Author Response

Many thanks for your comments and availability

Reviewer 2 Report (New Reviewer)

Dear Author,        

Thank you for your work. Plant breeding is a large project along with challenging for all researchers, especially in screening potential lines for further breeding projects. It is a novel concept to screen potential resources from commercial lines, which saves time and is more efficient. Your work is excellent and applicable. 

I would recommend accepting after spell-checking and language improvement.  

Author Response

Before the submission at Agriculture we send our paper to spell-checking and language improvement. More information can be found at the attached file  

Reviewer 3 Report (New Reviewer)

This study conducted an incomplete diallel cross between two groups (15x5) of rice materials, and gca and sca were analyzed. Overall, this study provided a preliminary genetic analysis of several traits in the rice hybrid diallel cross population. But I do not understand how the field experiment of F2 crosses were replicated with triple lattice design. As the experiment was performed in only one environment, inferences are limited. In addition, the primary purpose of this study is not clear, as well as many details of experiments and statistical analysis.

The following minor points are for the author’s reference.

L1: In my opinion, the title is too general.

L15: delete "highly".

L17: "significant" is not clear, do you mean variance or effect? are all effects significant?

L21: keywords are not well selected. "non-additive gene effects" is not clear.

L90: how F2 plant is selected from different segregation?

L94: F2 is heterozygous, how replicated?

L119: all effects should be clarified fixed or random.

L132: I suggest using "i.i.d." to represent independent and identically distributed, and N() for normal distribution.

L139-140: it is not clear how the quadratic component was estimated.

L146: how to interpret CVr, is it equal to the ratio of sample and genetic s.d.?

L162: how the parameters were estimated, more details of the analysis should be provided.

L163: statistical model and method should be provided.

L171: GCA1 and GCA2 should be clarified.

L193-194: ANOVA table should be provided, and trait heritability should be estimated.

Figure 1: The y-axis may be shared across subplots.

L255: ANOVA table should be provided.

L467: typo "CGA"

Author Response

"significant" is not clear, do you mean variance or effect? are all effects significant?

Here we are talking about the GCA and SCA effect, as presented in the Figure 1, except to GCA1 for Bs all the effects of GCA and SCA were significant.

L21: keywords are not well selected. "non-additive gene effects" is not clear.

We are not sure if we understand your question, because the keywords are a tool that helps indexers to find relevant papers. So we believe that “non-additive gene effects” is a good keywords which can facilitate our paper to be found by the large number of researches

L90: how F2 plant is selected from different segregation?

We didn’t select F2 plants, here we just talking that we self-pollinated F1 plants (crosses) to obtain the 25 F2 crosses

L94: F2 is heterozygous, how replicated?

We harvested all the panicles for each genotype (F1 plants), and we sown just 60 seeds per meter. So we had sufficiently number of the seeds to perform a trial with replicates

L119: all effects should be clarified fixed or random

We are not sure if we understand your question, because as described at line 119 all the effects of the model were considered as fixed effect

L132: I suggest using "i.i.d." to represent independent and identically distributed, and N() for normal distribution.

Ok, you are reason, we corrected.

L139-140: it is not clear how the quadratic component was estimated.

Ok, we tried to clarify the expression presented

L146: how to interpret CVr, is it equal to the ratio of sample and genetic s.d.?

The relative variation coefficient (CVr) also reflected situations of high precision favorable to selection, when CVr values are equal or larger than 1.0, suggests greater genetic variability, indicating that the selection is highly favorable

L162: how the parameters were estimated, more details of the analysis should be provided.

We are not sure if we understand your question, once time we didn’t estimate new parameters, we just performed a Tukey test to found differences between the groups established

L163: statistical model and method should be provided.

We think it’s not necessary to provide the Tukey statistical model, once time is a statistical method well established and studied

L171: GCA1 and GCA2 should be clarified.

As described at line 127 ( - parental group of commercial rice cultivars;  - parental group of 15 progenies of the CNA 12T population), so GCA1 and GCA2 is related to effects of this groups

L193-194: ANOVA table should be provided, and trait heritability should be estimated.

In the Figure 1 are presented a summary of the ANOVA table, and in the table 2 we provided the heritability estimates for each trait and groups

Figure 1: The y-axis may be shared across subplots.

As the x-axis have different scale, we choose not share the y-axis across subplots

255: ANOVA table should be provided.

Is provided at Figure 1

L467: typo "CGA"

You are reason, we corrected

This manuscript is a resubmission of an earlier submission. The following is a list of the peer review reports and author responses from that submission.

Round 1

Reviewer 1 Report

The article was written by Guimarães et al. titled 'Diallel analysis: choosing parents to introduce new variability in a recurrent selection population'. The current article describes the selection of appropriate donors, as well as information about the genetic basis of inheritance, which is important for exploitation by breeding programs using combing ability analysis.

However, combining ability analysis (??? and ???) for selecting parents for crossing is a very common study in plant breeding. Thus, this is incremental research (no novelty observed) that deals with established ideas and knowledge. 

Author Response

The article was written by Guimarães et al. titled 'Diallel analysis: choosing parents to introduce new variability in a recurrent selection population'. The current article describes the selection of appropriate donors, as well as information about the genetic basis of inheritance, which is important for exploitation by breeding programs using combing ability analysis.

However, combining ability analysis (??? and ???) for selecting parents for crossing is a very common study in plant breeding. Thus, this is incremental research (no novelty observed) that deals with established ideas and knowledge. 

In the strict sense we wanted to use the diallel analysis just to describes the selection of appropriate donors to introduce new variability in CNA 12T, an important population in Embrapa's rice breeding program to generate new lines with resistance to blast. As well as, we would like to get information about the genetic basis of the studied traits in this paper. In this case, we run a classical diallel analysis without introduce the molecular markers information, so we don't have the molecular information of the genotypes used for us in this study, there is the reason which this information was not introduced

Reviewer 2 Report

Parental matching is the key to rice breeding. This article reveals that wide diversity for different rice traits in the population,highly significant differences among parents and its crosses for all the traits, variance was highly significant for most of the traits studied in F2 progenies. The research results have certain theoretical significance and practical value for rice breeding.  The analysis process is comprehensive, good organized, large amount of information and so on. However, there are some major issues need to be improved:

  1. Abstract: Need to be modified and improved;
  2. Background: The latest literature on especially molecular markers was supplemented.
  3. Material and Methods:Diallel analysis is a traditional and classical method of genetic analysis; The genetic study of rice phenotype and molecular markers is remarkable. This is the insufficient part of the paper.
  4. Results:Keep title and tables on the same page as much as possible
  5. Discussion: The progress of rice molecular designfor diallel analysis  combined with the phenotype for breeding to make up for the shortcomings and highlight the innovation points of the paper.
  6. References:Some old references and their low correlation were removed.

Author Response

Parental matching is the key to rice breeding. This article reveals that wide diversity for different rice traits in the population,highly significant differences among parents and its crosses for all the traits, variance was highly significant for most of the traits studied in F2 progenies. The research results have certain theoretical significance and practical value for rice breeding.  The analysis process is comprehensive, good organized, large amount of information and so on. However, there are some major issues need to be improved:

Abstract: Need to be modified and improved;

Background: The latest literature on especially molecular markers was supplemented.

Yes, the abstract was not sufficiently and clearly developed. We shortened the text too much, so now we improved the section

Material and Methods: Diallel analysis is a traditional and classical method of genetic analysis; The genetic study of rice phenotype and molecular markers is remarkable. This is the insufficient part of the paper.

In the strict sense we wanted to use the diallel analysis just to describes the selection of appropriate donors to introduce new variability in CNA 12T, an important population in Embrapa's rice breeding program to generate new lines with resistance to blast. As well as, we would like to get information about the genetic basis of the studied traits in this paper. In this case, we run a classical diallel analysis without introduce the molecular markers information, so we don't have the molecular information of the genotypes used for us in this study, there is the reason which this information was not introduced

Results: Keep title and tables on the same page as much as possible

As possible we put the tables and titles in the same page

Discussion: The progress of rice molecular design for diallel analysis combined with the phenotype for breeding to make up for the shortcomings and highlight the innovation points of the paper.

The answer is given in the comment discussed above

References: Some old references and their low correlation were removed.

We revised the references, how the others reviewer considered the cited references were relevant to the research, we opted to did not delete any reference

Reviewer 3 Report

The manuscript presents the data on the performed diallel analysis of two genetic groups of rice for seven analyzed traits. Extensive statistical processing was performed. Although there are questions about the design of the experiment, the results may be of interest to breeders working with this economically important crop. Additional information should be added about how the 15 progeny of the S0:3 population of second-cycle CNA 12T rice selected as the female parent differed. It is also unclear how the crossing was done. Was a mixture of pollen from several plants of the same variety used? 
The conclusion "This study revealed wide diversity in various traits of rice in the analyzed population, but none of the 20 parents analyzed stood out as a good combinatorial parent simultaneously" may be related to the fact that the paternal component of the cross was not linear material and represented a varietal population. 
for detecting genetic patterns in the inheritance of traits and performing a diallel analysis, it is better to use lines.

Author Response

Additional information should be added about how the 15 progeny of the S0:3 population of second-cycle CNA 12T rice selected as the female parent differed. It is also unclear how the crossing was done. Was a mixture of pollen from several plants of the same variety used?

We revised the text, we shortened the text too much, so now we improved the explanation about how the crosses was done.

The conclusion "This study revealed wide diversity in various traits of rice in the analyzed population, but none of the 20 parents analyzed stood out as a good combinatorial parent simultaneously" may be related to the fact that the paternal component of the cross was not linear material and represented a varietal population for detecting genetic patterns in the inheritance of traits and performing a diallel analysis, it is better to use lines.

We understood your doubt, so now we revised the text and improved our conclusion

Round 2

Reviewer 1 Report

In keywords please include 'Griffing approach/method' instead of Griffing only.